# 3-Aryl-5-aminobiphenyl Substituted [1,2,4]triazolo[4,3-*c*]quinazolines: Synthesis and Photophysical Properties

**DOI:** 10.3390/molecules28041937

**Published:** 2023-02-17

**Authors:** Alexandra E. Kopotilova, Tatyana N. Moshkina, Emiliya V. Nosova, Galina N. Lipunova, Ekaterina S. Starnovskaya, Dmitry S. Kopchuk, Grigory A. Kim, Vasiliy S. Gaviko, Pavel A. Slepukhin, Valery N. Charushin

**Affiliations:** 1Department of Organic and Biomolecular Chemistry, Ural Federal University, 620002 Ekaterinburg, Russia; 2I. Postovsky Institute of Organic Synthesis, Ural Branch of the Russian Academy of Sciences, 620108 Ekaterinburg, Russia; 3M.N. Mikheev Institute of Metal Physics, Ural Branch of the Russian Academy of Sciences, 620108 Ekaterinburg, Russia

**Keywords:** [1,2,4]triazolo[4,3-*c*]quinazolines, cross-coupling, fluorescence, solvatochromism, pH-sensor

## Abstract

Amino-[1,1′]-biphenyl-containing 3-aryl-[1,2,4]triazolo[4,3-*c*]quinazoline derivatives with fluorescent properties have been designed and synthesized. The type of annelation of the triazole ring to the pyrimidine one has been unambiguously confirmed by means of an X-ray diffraction (XRD) method; the molecules are non-planar, and the aryl substituents form the pincer-like conformation. The UV/Vis and photoluminescent properties of target compounds were investigated in two solvents of different polarities and in a solid state. The samples emit a broad range of wavelengths and display fluorescent quantum yields of up to 94% in toluene solutions. 5-(4’-Diphenylamino-[1,1′]-biphenyl-4-yl)-3-(4-(trifluoromethyl)phenyl)-[1,2,4]triazolo[4,3-*c*]quinazoline exhibits the strongest emission in toluene and a solid state. Additionally, the solvatochromic properties were studied for the substituted [1,2,4]triazolo[4,3-*c*]quinazolines. Moreover, the changes in absorption and emission spectra have been demonstrated upon the addition of water to MeCN solutions, which confirms aggregate formation, and some samples were found to exhibit aggregation-induced emission enhancement. Further, the ability of triazoloquinazolines to detect trifluoroacetic acid has been analyzed; the presence of TFA induces changes in both absorption and emission spectra, and acidochromic behavvior was observed for some triazoloquinazoline compounds. Finally, electronic-structure calculations with the use of quantum-chemistry methods were performed for synthesized compounds.

## 1. Introduction

Heterocyclic scaffolds containing the triazole rings, which are annelated with natural pyrimidine heterocycles, represent “lead compounds” for organic synthesis [1], medicinal chemistry [2] and pharmacology [3,4]. Recently, the ability of some [1,2,4]triazolo[4,3-*c*]quinazolines to act as penetrating DNA intercalators has been demonstrated [5,6]. 3,5-Diphenyl[1,2,4]triazolo[4,3-*c*]quinazoline was reported as a selective A_3_A adenosine receptor antagonist [7], and 5-aryl-[1,2,4]triazolo[4,3-*c*]quinazoline-3-amine demonstrated anticonvulsant activity [8]. 

The main advantage of the azolodiazine domain of 1,2,4-triazole-containing hybrid molecules is associated with a planarized skeleton of aza-heterocyclic core, which has an important effect on structural modifications [9], biological interactions [10] and chromophore properties [11]. Benzoimidazo[1,2-*a*][1,2,3]triazolo[4,5-*e*]pyrimidines **A** (Figure 1) demonstrated aggregation-induced emissions (AIE) and acidochromic behaviour [12]. The emission spectra of 2*H*-[1,2,3]triazolo[4,5-e][1,2,4]triazolo[1,5-*a*]pyrimidine derivatives **B** in acetonitrile solutions are sensitive to the presence of nitroaromatic explosives [13]. Fluorescent 2-aryl-1,2,3-triazolopyrimidine **C** was demonstrated to penetrate cells and selectively accumulate in the cell membrane, Golgi complex and endoplasmic reticulum, which opens up wide opportunities in bioimaging [11].

The quinazolinyl moiety possesses a stronger electron-withdrawing ability than the pyrimidinyl core due to extra electron delocalization into the fused benzene ring. Enhanced intramolecular charge transfer (ICT) in quinazoline chromophores was illustrated by red-shifted absorption and emission with regard to their pyrimidine analogs [14]. There is limited data on the optical properties of 3,5-diarylsubstituted [1,2,4]triazolo[4,3-*c*]quinazolines. 5-Fluorenyl-, 5-spirofluorenyl and 5-(3(4)-fluorenyl)phenyl substituted 3-(4-cyanophenyl)-[1,2,4]triazolo[4,3-*c*]quinazolines (compounds **D**–**F**, Figure 2) were reported as novel electron transport heterocycles for efficient OLED materials possessing proper molecular dipole moments, which are able to be in compact contact with other organic layers in the process of forming a device [15]. Analogs of compounds **D**–**F** bearing a triphenyl-1,3,5-triazine fragment instead of a cyanophenyl one were mentioned in patent [16], and some 5-phenyl-, 5-biphenyl- and 5-napthyl- derivatives of [1,2,4]triazolo[4,3-*c*]quinazolines containing triphenyl-1,3,5-triazine residue in position C(3) have been developed [17].

Our research group is working on synthetic approaches to quinazoline fluorophores and investigating the photophysical properties, as well as possible practical applications. Recently, the π-conjugated chromophores based on 2-aryl/thienyl-4-cyanoquinazolines, including 2-(4′-amino[1,1′-biphenyl]-4-yl)-containing derivatives **G**–**I** (Figure 2), have been studied [18,19]. 4-Cyanoquinazolines are shown to be less emissive in solvents and solid state than quinazolin-4-ones [19] and morpholinyl [20] counterparts; the introduction of the cyano group into position C(4) leads to orange/red coloured powder and dual emission bands [19]. Annelation of the triazole cycle can be considered another way to enhance the electron-acceptor character of the quinazoline core.

Herein, we develop the series of [1,2,4]triazolo[4,3-*c*]quinazolines bearing 3-aryl and 5-aminobiphenyl fragments to handle the detailed investigation of photophysical properties and to discover preferable structures for practical applications. The synthetic approach involves the cross-coupling of a bromo-derivative with boronic acid under typical conditions. The target compounds appeared to exhibit fluorescent properties both in solution and solid state; the fluorophores demonstrated changes in absorption and emission spectra upon the addition of water to MeCN solutions. Moreover, triazoloquinazolines show sensing properties toward acid.

## 2. Results

### 2.1. Synthesis 

Triazoloquinazoline fluorophores **2a**–**i** were obtained by a Suzuki–Miyaura cross-coupling reaction. The approach involves a Pd-catalyzed interaction of the corresponding 4-bromophenyl derivative **1a**–**c** with arylboronic acid or an arylboronic acid pinacol ester; the yields were from moderate to good (25–65%), Figure 1. The starting 3-(*p*-tolyl), 3-(4-methoxyphenyl) or 3-(4-trifluoromethyl) substituted 5-(4-bromophenyl)-[1,2,4]triazolo[4,3-*c*]quinazolines **1a**–**c** were prepared from appropriate hydrazones by oxidative cyclization with bromine in glacial acetic acid at room temperature, as previously described [21]. 

^1^H NMR, ^13^C NMR spectroscopy, mass spectrometry (Appendix A) and elemental analysis data confirmed the identity and purity of target compounds. ^13^C NMR data for sample **2i** has not been obtained due to its poor solubility in organic solvents, including DMSO-d_6_ under heating.

Single crystals of quinazolines **2a** and **2e** were obtained by a slow evaporation technique (MeCN and *n*-hexane/chloroform mixture, respectively, used as a solvent) and analyzed by an X-ray diffraction method (XRD) (Figure 3, Appendix A). According to XRD data, compound **2a** is crystallized in the centrosymmetric space group of the monoclinic system. The molecule is non-planar, and the aryl substituents form the pincer-like conformation (Figure 3). The mean bond lengths and angles are near expectations. The configuration of the nitrogen atom in the diethyl amino group is near to planar. The deviation of the N(34) atom from the plane C(31)C(35)C(37) is 0.099 Å. The C–N bond distances of the ethyl substituents at the amino moiety are varied within a range of 1.456–1.462 Å. The distance C(31)–N(34) of 1.387 Å is significantly shortened, and the conjugation between the N(34) atom and the aryl substituent is observed. The shortened π-π interactions between the heterocyclic part and π-donating NEt_2_-phenyl substituent are observed, the distance C(5)…C(29) [1 − x, y − 0.5, 1.5 − z] was found to be 3.274 Å, which is 0.13 Å less than sum of the V-d-W radii.

Compound **2e** is crystallized in the non-primitive centrosymmetric space group. As well as **2a**, molecule **2e** is non-planar, and the aryl substituents form the pincer-like conformation (Figure 3). The mean bond lengths and angles were also close to expectations. The three N–C bonds at the triarylamino group are almost in the same plane. The deviation of the N(34) atom from the C(31)C(35)C(39) plane is 0.10 Å, significant torsion angles were defined between all aryl substituents and the plane of the C–N bonds. The C–N bond distances of the triarylamino moiety are varied within the range of 1.409–1.443 Å. The T-shaped Csp^2^–H…N π-contacts take place with the participation of the nitrogen atom of the triazolo moiety. No shortened π-π interactions in the molecular packing were found.

An undoubted significance of the XRD data obtained for compounds **2a**, **e** is that the type of annelation of the triazole moiety to the quinazoline one has been unambiguously confirmed. Unfortunately, there is still conflicting information in the literature; in particular, some patents report the formation of the [4,3-*c*] isomer of triazoloquinazoline [15,17] from arylhydrazone, and others state the [1,5-*c*] isomer [16] in the presence of the same oxidant PhI(OAc)_2_. Evidence of structure, causes of different reactions, and Dimroth rearrangement conditions were not provided.

### 2.2. UV/Vis and Fluorescence Spectroscopy

The UV/Vis absorption and photoluminescence (PL) spectroscopic data for toluene and MeCN solutions of [1,2,4]triazolo[4,3-*c*]quinazoline fluorophores **2a**–**i** are presented in Table 1; the corresponding spectra are shown in Appendix A. The solutions of a c ≈ 10^−5^ M concentration were prepared for the experiments. 

Based on the nature of substituent R^1^ in the aryl fragment, we have outlined the grouping of [1,2,4]triazolo[4,3-*c*]quinazoline chromophores **2a**–**i** for the discussion of optical properties: Me-group (**2a**–**c**), MeO-group (**2d**–**f**) and CF_3_-group (**2e**–**h**); in each group of compounds, the substituent NR_2_ in the biphenyl fragment was NEt_2_, NPh_2_ or 9*H*-carbazol-9-yl. 

The positions of the longest wavelength absorption bands of Et_2_N and Ph_2_N-containing fluorophores **2a**, **b**, **d**, **e**, **g**, **h** are observed in the UV region: 360–381 nm in toluene and 350–371 nm in MeCN (Table 1). The carbazolyl-derivatives **2c**, **2f** and **2i** are characterized by a blue-shifted (about 20–40 nm in toluene and 30–50 nm in MeCN) absorption band with respect to their diethylamino or diphenylamino counterparts, which is due to the decrease in conjugation length caused by the rigid structure of the carbazolyl unit. Notably, the profile of absorption spectra and positions of maxima in toluene are similar for compounds **2c**, **2f** and **2i** and are not influenced by the nature of substituent R^1^ in the aryl fragment (Me, OMe or CF_3_), whereas the absorption band of CF_3_C_6_H_4_-containing triazoloquinazolines with a diethyl- or diphenyl-amino fragment (**2g** and **2h**, respectively) is bathochromically shifted with respect to the corresponding fluorophores **2a**,**d** and **2b**,**e** (Table 1). 

Triazoloquinazolines **2a**–**i** have fluorescence emission maxima at 412–502 nm in toluene and 530–640 nm in MeCN, with quantum yields up to 94% (Table 1). We considered the influence of substituents R and R^1^ on PL characteristics. For example, a red shift in emission maximum was observed in the order **2d**→**2a**→**2g** for diethylamino-containing **2a**, **2d** and **2g** both in toluene and MeCN solution (Appendix A), which can be explained by the decrease in electron-donating influence of the CH_3_ group compared to MeO (compounds **2d** and **2a**) and by the increase in electron-withdrawing ability of the aryltriazol fragment due to the replacement of CH_3_ with CF_3_ (compounds **2a** and **2g**). For this set of compounds (**2a**, **2d** and **2g**), we observed the attenuation of PL emission intensity when going from toluene to MeCN, which was probably caused by the reinforced charge transfer process in the excited state. On the contrary, the emission band of diphenylamino-containing fluorophores **2b**, **2e** and **2h** does not depend on the nature of the aryl unit at the triazole ring (Appendix A). The QY of (trifluoromethyl)phenyl-[1,2,4]triazolo[4,3-*c*]quinazoline **2h** appeared to be the strongest in both solvents (Φ_F_ = 94% in toluene), which clearly indicates that the trifluorophenyl group plays an essential role in the intense emission for NPh_2_-containing derivatives **2b**, **e**, **h**. Fluorophores **2c**, **2f** and **2i** bearing the carbazolyl unit display the emission maxima at 530, 550 and 560 nm in MeCN and show the same dependence of emission wavelength on aryl fragment nature, **2f**→**2c**→**2i** (Appendix A) as their diethylamino counterparts. It is worth noting that derivatives **2c** (R^1^ = CH_3_) and **2i** (R^1^ = CF_3_) display a bimodal emission band in non-polar solvent (toluene) with a peak in high energy region at 467 and 434 nm, respectively, and at low energy, wavelengths at 515 and 517 nm, whereas compound **2f** (R^1^ = OCH_3_) only shows the high energy band in toluene with a maximum at 412 nm (Appendix A). The bimodal emission can arise, for example, due to the existence of associates with solvents or aggregates. For the establishment of the exact reasons, additional studies will be conducted. 

Along each series of quinazolines (Me-group, MeO-group, and CF_3_-group), the carbazolyl-containing fluorophores **2c**, **2f** and **2i** demonstrate emissions in the highest energetic region in MeCN (λ_max_ = 550, 530 and 560 nm, Table 1, Appendix A), which agrees with the low electron-donating ability of the carbazol-9-yl unit. The introduction of NEt_2_ or NPh_2_ results in the bathochromically shifted emission band (compounds **2a**, **b**, **d**, **e**, **g**, **h**, Appendix A). 

The quinazoline-based fluorophores **2a**–**i** display an emission in the range of 416–512 nm in the solid state (powder), as shown in Table 1. The influence of the electron-donating substituent NR_2_ on the emission maximum is similar in each group of compounds; a blue shift is observed in the order NEt_2_→NPh_2_→carbazol-9-yl. The quantum yield increases in the order NEt_2_→NPh_2_, and obviously, the steric effect of a non-planar propeller-like configuration of the NPh_2_ group prevents molecules from packing via π-π stacking and quenching [22,23,24,25,26]. (Figure 4, Appendix A, Table 1). A very high quantum yield of the compound **2h** (>95%) should be noted.

It is interesting to compare the triazolo-containing fluorophores **2a**–**i** with their 4-morpholinylquinazoline, 4-cyanoquinazoline or quinazolin-4-one counterparts containing the same 4′-amino[1,1′-biphenyl]-4-yl substituents [19,20]. We can conclude that the absorption maximum is not significantly influenced by the nature of the quinazoline core in the set of triazoloquinazoline, 4-morpholinylquinazoline and quinazolin-4-one, whereas emission bands of compounds **2a**–**i** undergo **a** shift to longer wavelengths in all cases compared to their analogs. For example, compound **2a** demonstrates an emission maximum at 491 nm in toluene and 612 nm in MeCN, whereas 2-(4′-N,N-diethylamino[1,1′-biphenyl]-4-yl)quinazolin-4(3*H*)-one exhibits emissions with λ_max_ = 450 nm in toluene and 535 nm in MeCN [19], and 2-(4′-N,N-diethylamino[1,1′-biphenyl]-4-yl)-4-(morpholin-4-yl)quinazoline has λ_max_ = 443 nm in toluene and 554 nm in MeCN [20]), which confirms the reinforcement of interactions between the donor and acceptor part. Moreover, photophysical characteristics of **2a**–**i** differ considerably from those of cyano derivatives (for example, the absence of a shoulder in the red region of the absorption band, a higher quantum yield both in solution and solid state, etc. [19]). Notably, in the series of triazoloquinazolines **2**, the great bathochromic shift of the emission band is observed when the electron donor NR_2_ group changes from NEt_2_ to NPh_2_ and further to carbazolyl (78–87 nm, Table 1), while for other types of fluorophores, the position of emission maximum is less dependent on NR_2_. This is probably because the presence of an aryl fragment in the triazole ring and the formation of a pincer-type structure prevents structure twisting, which leads to a more considerable and dependent correlation between the aryl donor group and emission maximum. 

### 2.3. Effects of Solvent Polarity for ***2a***, ***d***, ***g***, ***h***

A study of luminescent properties in solvents of different polarity was performed for compounds **2a**, **2d**, **2g** and **2h** possessing high emission intensity in toluene (Φ_F_ 47−94%) and MeCN (Φ_F_ 24−34%) (Appendix A). The values of the emission maxima in different solvents are presented in Table 2. It has been shown that with increasing solvent polarity, the emission maximum of these chromophores underwent a red shift; the positive emission solvatochromism was observed. The most significant changes in the emission spectrum were noticed for compound **2a** (λ_em_ = 451 nm in cyclohexane and λ_em_ = 648 nm in methanol) (Figure 5a). 

We observed a rather large difference in the emission maxima upon passing from the least polar to the most polar solvent for other studied compounds **2d**, **g**, **h** as well (Appendix A). 

In addition, a mathematical analysis of the solvatochromic behavior was performed for the obtained compounds according to the typical procedure based on the Lippert–Mataga equation [27,28,29]. The data are presented in Table 3 and Figure 5b. The linearity of the plots was found to confirm the positive solvatochromic effect in all the cases. The obtained values of the difference between the dipole moments of the ground and excited states are in the range of 17.54–22.34 D, and the maximum value of 22.34 D is observed for [1,2,4]triazolo[4,3-*c*]quinazoline **2h** bearing a triphenylamine donor moiety and *p*-(trifluoromethyl)phenyl residue. Notably, the nature of the aryl substituent at the triazole ring has a significant impact on the Δµ value (the value increases in the order **2d**→**2a**→**2g**). Thus, a pronounced intramolecular charge transfer (ICT) process upon photoexcitation occurs in molecules **2a**, **d**, **g**, **h**.

We analyzed the dependence of the emission maxima bathochromic shift on the quantitative polarity criteria of the solvents used, in particular, according to Kosower [31,32] and Dimroth/Reichardt mathematical models [33,34]. In most cases, the longest wavelength emission maximum is observed for methanol, and the shortest value corresponds to cyclohexane. In general, the obtained values of the emission maxima correspond to the criteria for the polarity of solvents. Deviations occur for compound **2d** bearing the 4-methoxyphenyl substituent at position 3; in this case, the longest wavelength maximum is characteristic of DMSO, but a very small difference between the values of emission maxima in DMSO and methanol was noticed (2 nm). However, in all cases, the emission maximum in DMSO corresponds with the longer wavelength than in the case of acetonitrile (the difference is 3–24 nm), even though the difference between their polarity criteria is minimal.

The comparison of the obtained characteristics of **2a**, **2d**, **2g**, **2h** with the ones of some early published (hetero)aryl-substituted quinazolines allowed us to conclude that the condensation of an additional triazole ring led to a significant increase in Δµ values. Thus, in the case of 2-(hetero)aryl-4-(4-aminophenyl)quinazolines [35], the obtained Δµ values were 15.93–18.26 D. At the same time, it should be noted that the introduction of the aromatic substituent, instead of the heteroaromatic one at position C(2) of quinazoline, led to an increase in Δµ values. For the group **2** compounds bearing aromatic substituents at the analogs position C(5), developed in the current work, the larger Δµ values (up to 22.34 D) were determined.

It should be noted that the revealed positive solvatochromism phenomenon opens up a number of prospects for the application of novel group **2** compounds as potential candidates for designing fluorescent probes and as components for fluorescent and non-linear optical materials [36]. 

### 2.4. Absorption and Fluorescence Behavior of Compounds ***2b***, ***2e*** and ***2h*** in MeCN/water Mixture

Frequently, triphenylamino (TPA)-containing compounds demonstrate enhancement of emission intensity (AIEE) or inducement of emission (AIE) upon aggregation [22,23,24,25,26]. The X-ray data (Figure 3) obtained for quinazoline derivative **2e** bearing the TPA unit confirms the non-planar structure of the studied molecules. We considered the absorption and emission behavior of the compounds **2b**, **2e** and **2h** upon the addition of water based on the above-mentioned analysis on passing from MeCN solution to solid state. For this aim, we registered absorption and emission spectra of fluorophores **2b**, **2e** and **2h** (at c = 2 × 10^−6^ M) in pure MeCN and in MeCN/water mixtures with various water fractions (f_w_); see Figure 6 and Appendix A. Figure 6 demonstrates changes in fluorescence spectra of **2e** in MeCN and MeCN/water mixtures (a) and a plot of relative PL intensity (I/I_0_) and wavelength at emission maxima of **2e** versus the composition of the water fraction (b) at room temperature. After the first portion of H_2_O, we observe a dramatic attenuation of emission intensity and red shift of the band. After 70vol% of water, the band shifts toward the blue region with the regeneration of intensity, and the strongest emission was measured at 80vol% of water. Other compounds (**2b** and **2h**) demonstrate similar changes in emission spectra. These changes are typical for luminogens with a rotatable donor–acceptor (D–A) structure, which possess two excited states: a locally excited (LE) state and a twisted intramolecular charge transfer (TICT) state. When the polarity of media increases (upon the addition of water), the TICT state becomes dominant, leading to red-shifted emission and the quenching of intensity. Further addition of water most likely led to the formation of aggregates that are less polar than initial single molecules, and this phenomenon resulted in blue-shifted emission. The intensity restores due to the restriction of rotation in the TPA group and phenylene moieties and the reducing non-radiative transitions. Thus, compounds **2b**, **2e** and **2h** exhibited aggregation-induced emission enhancement to some extent.

As for UV/vis spectra for quinazolines **2b**, **2e** and **2h**, we observed that the absorption band registered at 90%, 80% and 70% water fraction, respectively, was bathochromically shifted compared to pure MeCN, which can be ascribed to the formation of J-type aggregates with head-to-tail arrangement (Appendix A) [37,38,39]. 

Additionally, we carried out a time-resolved fluorescence study of quinazolines **2b**, **2e** and **2h** in a pure MeCN and MeCN/water mixture (Appendix A); fluorescence lifetimes were estimated to complete the photophysical characterization of the fluorophores. The lifetime of compound **2b** is fitted with bi-exponential decay with an average lifetime of 2.38 ns, whereas compounds **2e** and **2h** are characterized by mono-exponential decay curves and lifetimes of 3.17 and 2.46 ns, respectively (Appendix A). Fluorescence decay profiles of aggregated quinazolines at the highest emission intensity are fitted with three-exponential decay. Notably, the highest fractional contribution is different for all three compounds. While for *p*-tolyl-containing fluorophores, the highest fractional contribution at τ_2_ = 2.96 ns was detected, for their methoxy counterparts, it was observed at τ_1_ = 7.76 ns. On the contrary, the introduction of an electron-withdrawing CF_3_ group is reflected in the increased contribution of shorter time, τ_3_ = 1.36 ns (Appendix A). Generally, the average lifetimes of compounds **2b** and **2e** increase (for **2e** more considerable) and were found to be 3.57 and 7.72 ns, respectively, while τ_av_ of **2h** decreases by 1.36 ns in the MeCN/water mixture with respect to pure MeCN. 

### 2.5. Absorption and Fluorescence Behavior of Compounds ***2e*** and ***2h*** in Acidic Media

Because triazoloquinazolines **2a**–**i** bear nitrogen atoms that can bind with proton, we intend to investigate the absorption and fluorescence sensory properties of some samples toward acid. For the experiment, we have chosen quinazoline **2e**, containing an electron-donating *p*-methoxy substituent that reinforces the basicity of the triazole ring. On the contrary, the *p*-(trifluoromethyl)phenyl residue (compound **2h**) attenuates the basicity of the nitrogen atoms, and this derivative possesses the strongest fluorescence intensity in pure toluene, as was mentioned above. Nevertheless, both compounds demonstrate similar behavior upon the addition of TFA to toluene solution under UV light and daylight by the naked eye (Appendix A). The color of the solution turns from colorless to yellow (for **2e**) or orange (for **2h**) under daylight. The emission changes from dark blue or blue to orange. Both compounds demonstrate the reversibility of absorption and emission upon the consecutive addition of TFA and TEA. The titration experiment shows that the longwave absorption peak is red-shifted (from 360 to 372 nm in the case of **2e** and from 381 to 389 nm for **2h**), forming a tail-shaped band in the region of 380–440 nm and 420–460 nm, respectively, upon the addition of an excess of TFA (Figure 7a and Appendix A). The emission behaviour of fluorophore **2e** is more interesting in acidic media, and the changes are observed at a lower equivalent than that of compound **2h** (Figure 7). After the first portion of acid, we observe the bathochromic shift of the emission band and gradual enhancement of emission intensity. When 200 equivalents of TFA were added, the fluorescent intensity reached the highest value, and the peak at 528 nm appeared. The subsequent addition of acid resulted in the attenuation of intensity and further red shift by 540 nm at 500 eq of acid. 

The emission band of compound **2h** shifts to the red region with a gradual attenuation of intensity (Appendix A). The appearance of red-shifted absorption and emission bands in both cases can be associated with the reinforcement of electron-withdrawing strength of the triazoloquinazoline core due to the protonation of the nitrogen atom resulting in a stronger interaction of acceptor and donor units compared to neutral molecules. Therefore, significant acidochromic behavior was observed for compounds **2e**, **h**. We can suppose that changes in emission intensity (enhancement or quenching) of compounds **2e** and **2h** might be due to intramolecular photo-induced electron transfer (PET) or photo-induced proton transfer [40], and the elucidation of the exact interaction mechanism of the analyte with TFA is under progress. 

### 2.6. Quantum-Chemical Calculations

Furthermore, we performed the DFT calculations of quinazolines **2a**–**i** in the gas phase at the B3LYP/6–311 G* level using the Orca 4.0.1 software package [41,42,43,44,45] and conducted the chemical optimization on their energy levels based on DFT/B3LYP/6-31G (d,p) using Gaussian 09. The distribution plots of the HOMOs and LUMOs, as well as energy levels and energy gaps, are presented in Figure 7. Notably, the distribution plots of the HOMOs of diethylamino-containing fluorophores **2a** and **2d** are distinctly different from other counterparts. For molecules **2a** and **2d**, the HOMO electrons are mainly distributed on the 3-aryltriazolo[4,3-*c*]quinazoline fragment, which shows the considerable influence of the electronic nature of the substituent at the para-position of the phenylene ring. For compounds **2b**, **c**, **e**–**i,** the HOMO electrons are mainly located at the electron-donating arylamino unit, with the phenylene ring less involved in carbazolyl-derivatives **2c**, **2f** and **2i**, which confirms weaker π-conjugation of these molecules due to twisting of the rigid carbazolyl fragment and is consistent with photophysical data. The LUMOs plots are similar for all the compounds **2a**–**i** (Figure 8); electrons are distributed on the 5-(biphenyl-4-yl)-[1,2,4]triazolo[4,3-*c*]quinazoline framework. In general, the value of the energy gap decreases in each set of compounds bearing the same aminoaryl group with an increase in electron-withdrawing ability of the aryl substituent at position C(3). It is worth noting that energy levels of triazolo[4,3-*c*]quinazolines are closer to that of previously described 2-biphenylquinazolin-4(3*H*)-one derivatives than to ones for 4-morpholinyl or 4-cyano counterparts [19,20].

Obviously, different electronic distributions on HOMO and LUMO levels for compounds **2a**–**i** confirm the intense intramolecular charge transfer (ICT) process, which is consistent with solvatochromic studies for the considered compounds.

## 3. Experimental Methods

### 3.1. General Information

Unless otherwise indicated, all common reagents and solvents were used from commercial suppliers without further purification. Melting points were determined on Boetius-combined heating stages. ^1^H NMR and ^13^C NMR spectra were recorded at room temperature at 400 and 100 MHz, respectively, on a Bruker DRX-400 spectrometer (Bruker, Rheinstetten, Germany). Hydrogen chemical shifts were referenced to the hydrogen resonance of the corresponding solvent (DMSO-d_6_, δ = 2.50 ppm or CDCl_3_, δ = 7.26 ppm). Carbon chemical shifts were referenced to the carbon resonances of the solvent (DMSO-d_6_, δ = 39.5 ppm CDCl_3_, δ = 77.2 ppm). Peaks are labelled as singlet (s), doublet (d), triplet (t), quartet (q) and multiplet (m). Mass spectra were recorded on the SHIMADZU GCMS-QP2010 Ultra instrument (Shimadzu, Duisburg, Germany) with the electron ionization (EI) of the sample. Microanalyses (C, H, N) were performed using the Perkin–Elmer 2400 elemental analyser (Perkin–Elmer, Waltham, MA, USA).

### 3.2. Photophysical Characterization

UV/vis absorption spectra were recorded on the Shimadzu UV-1800 Spectrophotometer (Shimadzu, Duisburg, Germany) using quartz cells with 1 cm path length at room temperature. Emission spectra were measured on the Horiba FluoroMax-4 (HORIBA Ltd., Kyoto, Japan) at room temperature using quartz cells with 1 cm path length. The fluorescence quantum yield of the target compounds in solution and solid state were measured by using the Integrating Sphere Quanta-φ of the Horiba-Fluoromax-4. Time-resolved fluorescence measurements were carried out using time-correlated single-photon counting (TCSPC) with a nanosecond LED (λ = 370 nm).

### 3.3. Crystallography

The single crystal (light yellow block of 0.38 × 0.24 × 0.20) of compound **2a** and the single crystal (yellow block of 0.43 × 0.35 × 0.28) of compound **2e** were used for X-ray analysis. Structural studies were performed using equipment available in the Collaborative Access Center “Testing Center of Nanotechnology and Advanced Materials” at the Mikheev Institute of Metal Physics, Ural Branch, Russian Academy of Sciences. The X-ray diffraction analysis was performed at room temperature on Rigaku OD XtaLAB Synergy-S diffractometer (Rigaku Oxford Diffraction, Tokyo, Japan). Using Olex2 [46], the structure was solved with the ShelXT structure solution program using Intrinsic Phasing and refined with the ShelXL [47] refinement package using full-matrix least squares minimization. All non-hydrogen atoms were refined in an anisotropic approximation; the H-atoms were placed in the calculated positions and refined isotropically in the “rider” model. Crystal data for **2a** C_32_H_29_N_5_, M = 483.60, monoclinic, a = 12.64210(10) Å, b = 11.96340(10) Å, c = 17.2021(2) Å, α = 90°, β = 103.4260(10)°, γ = 90°, V = 2530.59(4) Å3, space group P2_1_/c, Z  =  4, μ(Mo Kα) = 0.077 mm^−1^. On the angles 4.75 < 2Θ < 52.744°, 5154 reflections were measured, 3812 unique (R_int_ = 0.1062), which were used in all calculations. Goodness to fit at F^2^ 1.151; the final R_1_ = 0.1290, wR_2_ = 0.2736 (all data) and R_1_ = 0.0770, wR_2_ = 0.2018 (I > 2σ(I)). Largest diff. peak and hole 0.21 and –0.23 ēÅ^−3^.

Crystal data for **2e** C_40_H_29_N_5_, M = 595.68, orthorhombic, a = 76.1636(13) Å, b = 17.3954(3) Å, c = 9.57464(12) Å, α = 90°, β = 90°, γ = 90°, V = 12685.4(4) Å3, space group Fdd2, Z = 16, μ(Mo Kα) = 0.077 mm^−1^. On the angles 2.14 < 2Θ < 29.48°, 8203 reflections were measured, 3575 unique (R_int_ = 0.2091), which were used in all calculations. Goodness to fit at F^2^ 0.975; the final R_1_ = 0.1695, wR_2_ = 0.2165 (all data) and R_1_ = 0.0605, wR_2_ = 0.1491 (I > 2σ(I)). Largest diff. peak and hole 0.18 and –0.20 ēÅ^−3^.

The results of the X-ray diffraction analysis for compounds **2a** and **2e** were deposited with the Cambridge Crystallographic Data Centre (CCDC 2218321 for **2a** and CCDC 2217179 for **2e**). The data are free and can be accessed at www.ccdc.cam.ac.uk (accessed on 8 November 2022 for **2a** and 3 November 2022 for **2e**).

### 3.4. General Procedures of Suzuki Cross-Coupling

To the mixture of bromo derivative **1a**–**c** (0.65 mmol) in toluene (10 mL), the corresponding boronic acid or boronic acid pinacol ester (0.70 mmol), PdCl_2_(PPh_3_)_2_ (46 mg, 65 mol), PPh_3_ (34 mg, 130 μmol), saturated solution of K_2_CO_3_ (3.7 mL) and EtOH (3.7 mL) were added. The mixture was stirred at 85 °C for 7–20 h in argon atmosphere in a round-bottom pressure flask. The reaction mixture was cooled. After cooling, the organic layer was separated, washed sequentially with EtOAc (10 mL) and H_2_O (10 mL), dried over Na_2_SO_4_, and the organic layer was evaporated at reduced pressure. The product was isolated by gradient column chromatography on silica gel, a mixture of hexane and ethyl acetate was used as an eluent.

**5-(4′-Diethylamino-[1,1′]-biphenyl-4-yl)-3-(*p*-tolyl)-[1,2,4]triazolo[4,3-*c*]quinazoline (2a).** Yellow powder, yield 25%; mp 195–197 °C; ^1^H NMR (CDCl_3_, 400 MHz) δ 1.21 (6H, t, *^3^J* = 7.2 Hz, 2CH_3_), 2.24 (3H, s, CH_3_), 3.42 (4H, q, ^3^*J* = 7.2 Hz, 2CH_2_), 6.74 (2H, d, ^3^*J* = 8.3 Hz, 2CH_phenylene_), 6.91 (2H, d, ^3^*J* = 7.8 Hz, 2CH_phenylene_), 7.08 (2H, d, ^3^*J* = 7.8 Hz, 2CH_phenylene_), 7.23–7.30 (4H, m, 4CH_phenylene_), 7.36 (2H, d, ^3^*J* = 8.3 Hz, 2CH_phenylene_), 7.73 (1H, m, CH_quinaz_), 7.81 (1H, m, CH_quinaz_), 8.04 (1H, d, ^3^*J* = 8.1 Hz, m, CH_quinaz_), 8.76 (1H, d, ^3^*J* = 7.8 Hz, m, CH_quinaz_); ^13^C NMR (CDCl_3_, 100 MHz) δ 12.7 (2CH_3_), 21.4 (CH_3_), 44.5 (2CH_2_), 111.9, 116.5, 123.6, 125.0, 125.3, 126.7, 128.1, 128.4, 128.6, 129.1, 129.3, 129.5, 129.7, 131.8, 139.5, 141.3, 143.7, 146.1, 147.8, 149.3, 150.0; EIMS *m*/*z* 484 [M + 1]^+^ (26), 483 [M]^+^ (69), 469 (37), 468 (100) 234 (21); anal. C 79.47, H 14.48, N 8.94%, calcd for C_32_H_29_N_5_ (483.62) C 79.40, H 6.12, N 14.11%.

**5-(4′-Diphenylamino-[1,1′]-biphenyl-4-yl)-3-(*p*-tolyl)-[1,2,4]triazolo[4,3-*c*]quinazoline (2b).** Beige powder, yield 48%; mp 205–207 °C; ^1^H NMR (CDCl_3_, 400 MHz) δ 2.24 (3H, s, CH_3_), 6.93 (2H, d, ^3^*J* = 7.7 Hz, 2CH_phenylene_), 7.09–7.18 (10H, m), 7.28–7.36 (10H, m), 7.76 (1H, m, CH_quinaz_), 7.84 (1H, m, CH_quinaz_), 8.06 (1H, d, ^3^*J* = 8.0 Hz, CH_quinaz_), 8.78 (1H, d, ^3^*J* = 7.9 Hz, CH_quinaz_); ^13^C NMR (100 MHz, CDCl_3_) δ 21.4 (CH_3_), 116.6, 123.5, 123.6, 124.9, 125.0, 126.0, 127.9, 128.4, 128.6, 129.3, 129.4, 129.5, 129.7, 130.6, 131.9, 133.7, 139.5, 141.3, 143.1, 145.8, 147.6, 148.1, 149.2, 150.0; EIMS *m*/*z* 580 [M + 1]^+^ (44), 579 [M]^+^ (100), 578 (11), 289 (17); anal. C 82.88, H 5.04, N 12.08%, calcd for C_40_H_29_N_5_ (579.71) C 82.88, H 5.01, N 12.00%.

**5-(4′-(9*H*-Carbazol-9-yl)-[1,1′]-biphenyl-4-yl)-3-(*p*-tolyl)-[1,2,4]triazolo[4,3-*c*]quinazoline (2c).** After cooling, the reaction mixture was filtered and washed with hexane. Colourless powder, yield 65%; mp 236–238 °C; ^1^H NMR (CDCl_3_, 400 MHz) δ 2.27 (3H, s, CH_3_), 6.97 (2H, d, ^3^*J* = 7.8 Hz, 2CH_phenylene_), 7.12 (2H, d, ^3^*J* = 7.8 Hz, 2CH_phenylene_), 7.32 (2H, m, 2CH_carbaz_), 7.42–7.50 (10H, m), 7.69 (4H, m), 7.77 (1H, m, CH_quinaz_), 7.84 (1H, m, CH_quinaz_), 8.07 (1H, d, ^3^*J* = 8.1 Hz, CH_quinaz_), 8.17 (2H, d, ^3^*J* = 7.8 Hz, 2CH_carbaz_), 8.80 (1H, d, ^3^*J* = 7.8 Hz, CH_quinaz_); ^13^C NMR (100 MHz, CDCl_3_) δ 21.5 (CH_3_), 109.9, 116.6, 120.3, 120.6, 123.6, 123.7, 125.0, 126.2, 126.6, 127.6, 128.5, 128.6, 128.7, 129.4, 129.6, 129.8, 131.6, 132.0, 137.8, 139.2, 139.6, 140.9, 141.2, 142.6, 145.6, 149.1, 150.0; EIMS *m*/*z* 579 [M + 2]^+^ (10), 578 [M + 1]^+^ (45), 577 [M]^+^ (100), 344 (11), 288 (24), 241 (11), 102 (17), 88 (12), 77 (14), 57 (10), 44 (39), 43 (28), 41 (13); anal. C 83.17, H 4.71, N 12.12%, calcd for C_40_H_27_N_5_ (577.69.) C 83.12, H 4.11, N 12.15%.

**5-(4′-Diethylamino-[1,1′]-biphenyl-4-yl)-3-(4-methoxyphenyl)-[1,2,4]triazolo[4,3-*c*]quinazoline (2d).** After cooling, the reaction mixture was filtered and washed with hexane. The product was additionally recrystallized from CH_2_Cl_2_/hexane mixture. Yellow powder, yield 29%; mp 189–191 °C; ^1^H NMR (CDCl_3_, 400 MHz) δ 1.21 (6H, t, *^3^J* = 7.0 Hz, 2CH_3_), 3.41 (4H, q, ^3^*J* = 7.0 Hz, 2CH_2_), 3.63 (3H, s, OCH_3_), 6.62 (2H, d, ^3^*J* = 8.5 Hz, 2CH_phenylene_), 6.74 (2H, d, ^3^*J* = 8.5 Hz, 2CH_phenylene_), 7.11 (2H, d, ^3^*J* = 8.1 Hz, 2CH_phenylene_), 7.8 (4H, m, 4CH_phenylene_), 7.37 (2H, d, ^3^*J* = 8.1 Hz, 2CH_phenylene_), 7.73 (1H, m, CH_quinaz_), 7.78 (1H, m, CH_quinaz_), 8.03 (1H, d, ^3^*J* = 7.3, CH_quinaz_), 8.75 (1H, d, ^3^*J* = 7.2, CH_quinaz_); ^13^C NMR (100 MHz, CDCl_3_) 12.8 (2CH_3_), 44.8 (2CH_2_), 55.5 (OCH_3_), 112.0, 113.4, 116.6, 120.2, 123.5, 125.3, 126.7, 128.1, 128.4, 129.1, 129.4, 129.5, 131.2, 131.8, 141.3, 143.6, 146.1, 147.8, 149.1, 150.0, 160.6; EIMS *m*/*z* 500 [M + 1]^+^ (32), 499 [M]^+^ (85), 485 (34), 484 (100), 242 (47), 228 (11), 207 (10); anal. C 76.93, H 5.85, N 14.02%, calcd for C_32_H_29_N_5_O (499.62) C 76.90, H 5.87, N 14.10%.

**5-(4′-Diphenylamino-[1,1′]-biphenyl-4-yl)-3-(4-methoxyphenyl)-[1,2,4]triazolo[4,3-*c*]quinazoline (2e).** The product was additionally recrystallized from CH_2_Cl_2_/hexane mixture. Pale yellow powder, yield 51%; mp 242–244 °C; ^1^H NMR (CDCl_3_, 400 MHz) δ 3.64 (3H, s, OCH_3_), 6.62 (2H, d, ^3^*J* = 6.6 Hz, 2CH_phenylene_), 7.02–7.16 (10H, m), 7.26–7.36 (10H, m), 7.74 (1H, m, CH_quinaz_), 7.82 (1H, m, CH_quinaz_), 8.04 (1H, d, ^3^*J* = 8.2 Hz, CH_quinaz_), 8.76 (1H, d, ^3^*J* = 7.7 Hz, CH_quinaz_); ^13^C NMR (100 MHz, CDCl_3_) δ 55.4 (OCH_3_), 113.5, 116.6, 120.2, 123.5, 123.6, 124.8, 126.0, 127.9, 128.4, 129.3, 129.5, 129.5, 130.6, 131.2, 131.9, 133.7, 141.2, 143.0, 145.8, 147.6, 148.1, 149.0, 149.9, 160.6; EIMS *m*/*z* 597 [M + 2]^+^ (10), 596 [M + 1]^+^ (45), 595 [M]^+^ (100), 298 (20); anal. C 80.65, H 4.91, N 11.76%, calcd for C_40_H_29_N_5_O (595.71) C 89.90, H 4.77, N 11.44%.

**5-(4′-(9*H*-Carbazol-9-yl)-[1,1′]-biphenyl-4-yl)-3-(4-methoxyphenyl)-[1,2,4]triazolo[4,3-*c*]quinazoline (2f).** After cooling, the reaction mixture was filtered and washed with hexane. The product was additionally recrystallized from DMSO. Grey powder, yield 38%; mp 277−279 °C; ^1^H NMR (DMSO-*d*_6_, 400 MHz) δ 3.64 (3H, s, OCH_3_), 6.68 (2H, d, ^3^*J* = 8.5 Hz, 2CH_phenylene_), 7.17 (2H, d, ^3^*J* = 8.6 Hz, 2CH_phenylene_), 7.31 (2H, m, 2CH_carbaz_), 7.42 – 7.50 (m, 8H), 7.74 (2H, d, ^3^*J* = 8.2 Hz, 2CH_phenylene_), 7.80–7.86 (3H, m), 7.91 (1H, m, CH_quinaz_), 8.04 (1H, d, ^3^*J* = 8.0 Hz, CH_quinaz_), 8.24 (2H, m, 2CH_carbaz_), 8.63 (1H, d, ^3^*J* = 7.8 Hz, CH_quinaz_); ^13^C NMR (100 MHz, CDCl_3_) δ 55.4(OCH_3_), 109.9, 113.5, 116.6, 120.2, 120.3, 120.6, 123.6, 123.7, 126.2, 126.6, 127.6, 128.5, 128.7, 129.4, 129.7, 131.3, 131.6, 131.9, 137.8, 139.2, 140.9, 141.2, 142.5, 145.6, 148.9, 149.9, 160.6; EIMS *m/z* 594 [M + 1]^+^ (44), 593 [M]^+^ (100), 344 (16), 319 (14), 296 (30), 241 (17), 223 (12), 166 (17), 140 (12), 102 (12), 88 (17), 55 (16), 44 (32), 43 (38), 42 (19), 41 (11), 39 (17); anal. C 80.68, H 4.33, N 11.97%, calcd for C_40_H_29_N_5_O (593.69) C 80.92, H 4.58, N 11.80%.

**5-(4′-Diethylamino-[1,1′]-biphenyl-4-yl)-3-(4-(trifluoromethyl)phenyl)-[1,2,4]triazolo[4,3-*c*]quinazoline (2g).** The product was additionally recrystallized from CH_2_Cl_2_/hexane mixture. Yellow powder, yield 46%; mp 202–204 °C; ^1^H NMR (CDCl_3_, 400 MHz) δ 1.21 (6H, t, *^3^J* = 7.1 Hz, 2CH_3_), 3.41 (4H, q, ^3^*J* = 7.1 Hz, 2CH_2_), 6.72 (2H, d, ^3^*J* = 7.4 Hz, 2CH_phenylene_), 7.30–7.42 (10H, m), 7.76 (1H, m, CH_quinaz_), 7.84 (1H, m, CH_quinaz_), 8.06 (1H, d, ^3^*J* = 8.1 Hz, CH_quinaz_), 8.77 (1H, d, ^3^*J* = 7.9 Hz, CH_quinaz_); ^19^F NMR (CDCl_3_, 376 MHz) δ – 62.73 (3F, s, CF_3_); ^13^C NMR (100 MHz, CDCl_3_) δ 12.7 (2C, 2CH_3_), 44.5 (2C, 2CH_2_), 111.9, 116.3, 123.6 (1C, q, ^1^*J_CF_* = 272.5 Hz, CF_3_) 123.7, 124.7 (2C, q, ^3^*J_CF_* = 3.7 Hz, C_6_H_5_CF_3_), 125.6, 126.2, 128.1, 128.3, 128.5, 128.9, 129.2, 129.3, 129.4, 130.0, 130.2, 131.2, 131.5, 131.7, 132.2, 141.4, 144.5, 145.5, 147.9, 148.0, 150.6; EIMS *m*/*z* 538 [M + 1]^+^ (30), 537 [M]^+^ (78), 523 (36), 522 (100), 465 (12), 390 (11), 261 (25), 176 (10), 161 (12), 146 (12), 49 (12); anal. C 71.50, H 4.88, N 13.03%, calcd for C_32_H_26_F_3_N_5_ (537.59) C 71.40, H 4.98, N 13.00%.

**5-(4′-Diphenylamino-[1,1′]-biphenyl-4-yl)-3-(4-(trifluoromethyl)phenyl)-[1,2,4]triazolo[4,3-*c*]quinazoline (2h).** Yellow powder, yield 64%; mp 197–199 °C; ^1^H NMR (CDCl_3_, 400 MHz) δ 7.08 (2H, m, 2H_phenyl_), 7.14–7.21 (6H, m, 4H_phenyl_, 2H_phenylene_), 7.27–7.33 (4H, m, 4 CH_phenyl_), 7.61 (2H, d, ^3^*J* = 6.5 Hz, 2CH_phenylene_), 7.76 (1H, m, CH_quinaz_), 7.80 (2H, d, *J* = 8.1 Hz, 2H_phenylene_), 7.85 (2H, d, ^3^*J* = 8.0 Hz, 2H_phenylene_), 7.90 (1H, m, CH_quinaz_), 8.17 (1H, d, ^3^*J* = 8.1 Hz, CH_quinaz_), 8.56 (d, 2H, ^3^*J* = 8.0 Hz, 2H_phenylene_), 8.67 (1H, d, ^3^*J* = 7.9 Hz, CH_quinaz_), 8.75 (2H, d, ^3^*J* = 8.1 Hz, 2H_phenylene_); ^19^F NMR (CDCl_3_, 376 MHz) δ – 62.73 (3F, s, CF_3_); ^13^C NMR (100 MHz, CDCl_3_) δ 117.4, 123.5, 123.6, 123.9, 124.3 (1C, q, ^1^*J_CF_* = 271.7 Hz, CF_3_), 124.9, 125.8, 125.9, 126.6, 128.1 (2C, q, ^3^*J_CF_* = 3.7 Hz, C_6_H_5_CF_3_), 128.5, 129.0, 129.5, 130.0, 131.1, 132.1, 132.4, 133.6, 133.9, 143.2, 144.0, 146.3, 147.7, 148.2, 153.3, 162.8; EIMS *m*/*z* 635 [M + 2]^+^ (14), 634 [M + 1]^+^ (55), 633 [M]^+^ (100), 318 (10), 317 (34), 231 (14), 218 (13), 167 (21), 166 (15), 77 (19); anal. C 75.82, H 4.14, N 11.05%, calcd for C_40_H_26_F_3_N_5_ (633.68) C 75.72, H 4.18, N 10.98%.

**5-(4′-(9*H*-Carbazol-9-yl)-[1,1′]-biphenyl-4-yl)-3-(4-(trifluoromethyl)phenyl)-[1,2,4]triazolo[4,3-*c*]quinazoline (2i).** After cooling, the reaction mixture was filtered and washed with hexane. Grey powder, yield 50%; mp 305–307 °C; ^1^H NMR (CDCl_3_, 400 MHz) δ 7.32 (2H, m, 2CH_carbaz_), 7.39–7.49 (12H, m), 7.69 (4H, m), 7.81 (1H, m, CH_quinaz_), 7.88 (1H, t, CH_quinaz_), 8.10 (1H, d, ^3^*J* = 8.2 Hz, CH_quinaz_), 8.17 (2H, d, ^3^*J* = 7.9 Hz, 2CH_carbaz_), 8.81 (1H, d, ^3^*J* = 7.9 Hz, CH_quinaz_); ^19^F NMR (CDCl_3_, 376 MHz) δ – 61.29 (3F, s, CF_3_); EIMS *m*/*z* 633 [M + 2]^+^ (10), 632 [M + 1]^+^ (45), 631 [M]^+^ (100), 344 (16), 342 (11), 315 (34), 264 (12), 241 (19), 230 (24), 201 (11); anal. C 76.06, H 3.83, N 11.09%, calcd for C_40_H_24_F_3_N_5_ (631.66) C 76.04, H 3.80, N 11.15%.

## 4. Conclusions

Nine push-pull molecules based on [1,2,4]triazolo[4,3-*c*]quinazolines were designed and synthesized by cross-coupling reactions. The structure of target compounds, namely the arrangement of the triazole ring, has been confirmed by means of the X-ray data method. The triazoloquinazolines **2** were shown to emit in solution and solid state. Photophysical properties (absorption, emission and QY) are influenced by the nature of arylamino residue, 3-aryltriazole fragment, as well as solvent polarity. Variations of structure and media can be used for the fine-tuning of characteristics that are necessary for practical application. Moreover, fluorophores display solvatochromic behavior and their emission maxima show bathochromic shifts with an increase in solvent polarity. Furthermore, changes in absorption and emission spectra upon the addition of water to MeCN solution, attributed to the aggregation process, have been shown. The twisted structure of fluorophores revealed by X-ray analysis can inhibit intermolecular π-π stacking interactions, thus favoring the strong solid-state emission and active aggregate forms. Additionally, triazoloquinazolines display reversible changes in color and optical properties upon the treatment with TFA and have the potential application as sensors. In general, this type of fluorophore represents a promising group of compounds for different applications and further investigation.

## Data Availability

The data are available on request from the corresponding authors.

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
