# Peer review of "3-Aryl-5-aminobiphenyl Substituted [1,2,4]triazolo[4,3-c]quinazolines: Synthesis and Photophysical Properties"

_molecules, 2023, doi:10.3390/molecules28041937_

Round 1

Reviewer 1 Report

3-Aryl-5-aminobiphenyl-substituted [1,2,4]triazolo[4,3-c]quinazolines: Synthesis and Photophysical Properties by Kopotilova A. E. et al. presents a synthesis strategy for this interesting class of compounds of significant importance, namely the optical properties of these compounds and possible applications.

X-ray data presented herein is a valuable complement to NMR data for the investigation of potential isomerism of target compounds.

I recommend that this article should be published in the journal Molecules as it is well written and interesting. But on the condition that the following minor points are addressed:

Lines 81-82, please revise this ambiguous sentence.

Fiures 1-8, the quality / resolution must be dramatically improved

Line 173, please change "R" to "R1" as in Scheme 1.

Line 218, "absent", should be revised, i think "absence" is more appropriate

Lines 254 and 276, there should be additional space before references, for instance as in line 254, "equation[22-24] should be revised throughout manuscript"

Line 309, "adsorption", please correct to "absorption"

Line 310, "the dramatically", maybe "a dramatic" fits better?

Line 316, "that less", maybe "that are less" ?

Line 330, "compare" maybe "compared" ?

Regards referencing of NMR data, for the future publications, as per IUPAC recomendations you should use TMS for a greater reliability as the chemical shifts (δ), expressed in ppm, should be relative to tetramethylsilane (TMS).

Solvent residual peaks i would choose only as a secondary option.

Lines 538, 551 and 562, please correct the resonance frequency of 19F to 376 MHz.

For compounds with CF3, i would recommend to provide J coupling data in 13C NMR, you should see at least from 1J to 3J quartets.

The 19F NMR data looks good, the chemical shifts for CF3 looks good, no concerns regards this. The J coupling in the 13C NMR is just a complementary.

Supporting infromation:

1. Section.... [4,3-c], c is not in italics, please correct

The labels of figures seems strange, "a, B, c" why b in capital?

Please correct DCCCl3 to CDCl3 throughout suporting information.

The quality of pictures, especially of NMR data is quite bad quality, low resolution.

For instance if you copy/paste from Mnova your NMR data, i would suggest to paste special, Picture (enchanced metafile). Please improve it.

Overally, good quality work.

Best of luck.

Author Response

+ Lines 81-82, please revise this ambiguous sentence.

The sentence was revised as follow: The target compounds appeared to exhibit fluorescent properties both in solution and solid state; the fluorophores demonstrate changes in absorption and emission spectra upon addition of water to MeCN solutions. 

+ Figures 1-8, the quality / resolution must be dramatically improved

Probably, the bad resolution happen as a result of conversion of word file to PDF format. We saved all pictures of manuscript as distinct PDF files, also we saved SI in PDF format. All files are attached with this response. The resolution of pictures is rather good.

+ Line 173, please change "R" to "R1" as in Scheme 1.

We changed "R" to "R1".

+ Line 218, "absent", should be revised, i think "absence" is more appropriate

Sure, we corrected the word "absent" into "absence".

+ Lines 254 and 276, there should be additional space before references, for instance as in line 254, "equation[22-24] should be revised throughout manuscript"

We added space before reference in necessary places.

+ Line 309, "adsorption", please correct to "absorption"

We corrected the typo.

+ Line 310, "the dramatically", maybe "a dramatic" fits better?

It’s right, “a dramatic" fits better.

+ Line 316, "that less", maybe "that are less"?

"That are less" will be correct.

+ Line 330, "compare" maybe "compared" ?

We corrected "compare" into "compared".

+ Regards referencing of NMR data, for the future publications, as per IUPAC recomendations you should use TMS for a greater reliability as the chemical shifts (δ), expressed in ppm, should be relative to tetramethylsilane (TMS).

Solvent residual peaks i would choose only as a secondary option.

Thank you for this recommendation. We will take it into account for future publications.

+ Lines 538, 551 and 562, please correct the resonance frequency of 19F to 376 MHz.

The resonance frequency of 19F was corrected.

+ For compounds with CF3, i would recommend to provide J coupling data in 13C NMR, you should see at least from 1J to 3J quartets.

The 19F NMR data looks good, the chemical shifts for CF3 looks good, no concerns regards this. The J coupling in the 13C NMR is just a complementary.

We provided J coupling data in 13C NMR where it is possible. For example, for 2h: 13C NMR (100 MHz, CDCl3) δ 117.4, 123.5, 123.6, 123.9, 124.3 (1C, q, 1JCF = 271.7 Hz, CF3), 124.9, 125.8, 125.9, 126.6, 128.1 (2C, q, 3JCF = 3.7 Hz, C6H5CF3), 128.5, 129.0, 129.5, 130.0, 131.1, 132.1, 132.4, 133.6, 133.9, 143.2, 144.0, 146.3, 147.7, 148.2, 153.3, 162.8;

Supporting information:

+ Section.... [4,3-c], c is not in italics, please correct

We did “c” in italics.

+  The labels of figures seem strange, "a, B, c" why b in capital?

Probably, this mistake happens after conversion of word file to PDF format. We saved correct SI in PDF format and attached it.

+ Please correct DCCCl3 to CDCl3 throughout suporting information.

We corrected DCCl3 to CDCl3 throughout supporting information

+ The quality of pictures, especially of NMR data is quite bad quality, low resolution.

For instance if you copy/paste from Mnova your NMR data, i would suggest to paste special, Picture (enchanced metafile). Please improve it.

 Probably, the bad resolution happen as a result of conversion of word file to PDF format. We saved all pictures of manuscript as distinct PDF files, also we saved SI in PDF format. All files are attached with this response. The resolution of pictures is rather good.

Reviewer 2 Report

Limited number of molecules, what in my opinion can not be considered a SAR. Namely, the authors use three different groups in R1: The trifluoromethyl (-CF3) group is one of the most powerful electron withdrawing groups in structural organic chemistry. On the other hand, both CH3- and CH3O-  attached to a conjugated system- group will donate electron pair via inductive effect and hyperconjugation and resonance, respectively.  In this context, at least one weak electron-withdrawing group should be added to the SAR, to could correlate the photophysical results.

The characterization of the compounds is complete; despite no exact mass with four decimal places was presented, elemental analysis covers this gap.

Absorbance and emission studies completed, different solvents, different sates (liquid, solid), presence of water, acidic conditions

In summary is a good photophysical study but do not have enough SAR to take conclusions and some of them could be precipitated.

Author Response

Indeed, the discussion of structure-property relationship is not correct having only one type of electron acceptor substituent. Therefore, we rewrite several sentences in the article. For example: “Herein we develop the series of [1,2,4]triazolo[4,3-c]quinazolines bearing 3-aryl and 5-aminobiphenyl fragments to handle detailed investigation of photophysical properties and to discover preferable structures for practical applications.”

It is worth noting that the obtaining of another type of fluorophores with electron accepting group (4-cyanophenyl) failed.

Reviewer 3 Report

The manuscript titled "3-Aryl-5-aminobiphenyl substituted [1,2,4]triazolo[4,3- 2 c]quinazolines: synthesis and photophysical properties" is well written and can be accepted after minor corrections in English writing.

Author Response

We performed corrections in English writing across the article, we have rewritten some sentences, corrected style, and grammar issues.

Reviewer 4 Report

In this work, the authors designed and synthesized 3-aryl - [1,2,4] triazoline [4,3-c] quinazoline derivatives containing amino - [1,1 '] - biphenyl residue at the fifth position of heterocyclic nucleus, and studied their photoluminescence properties in two different polar solvents and solid states. These samples emit a wide wavelength range with fluorescence quantum yields up to 94% in the solution state. In addition, the solvent photochromic properties, optical properties such as AIE and detecting TFA properties are studied. The data support the conclusions. Based on my opinion, I suggest to accept this work after a major revision.

1. Some pictures (Fig. 3, 4, 5,6,7,8) are not clear enough.

2. In section 2.2, why 2c and 2i show bimodal emission bands in nonpolar solvent toluene is not explained.

3. In section 2.4, the explanation of Figure 6 (a) is not clear. In addition, it is impossible to distinguish whether the maximum fluorescence intensity is 50% water fraction or 80% water fraction, because the picture is not clear enough.

4. The author should use three-line form regarding to all the Tables.

5. The mechanization regarding to the detecting TPA should be provided.

Author Response

+ 1. Some pictures (Fig. 3, 4, 5,6,7,8) are not clear enough.

 Probably, the bad resolution happen as a result of conversion of word file to PDF format. We saved all pictures of manuscript as distinct PDF files, also we saved SI in PDF format. All files are attached with this response. The resolution of pictures is rather good.

+ 2. In section 2.2, why 2c and 2i show bimodal emission bands in nonpolar solvent toluene is not explained.

The possible reason of bimodal emission bands is formation associates with solvent (toluene) or aggregates. This phenomenon has been registered for described previously structures too [Moshkina, T.N.; Nosova, E.V.; Permyakova, J.V.; Lipunova, G.N.; Zhilina, E.F.; Kim, G.A.; Slepukhin, P.A.; Charushin, V.N. Push-Pull Structures Based on 2-Aryl/Thienyl Substituted Quinazolin-4(3H)-Ones and 4-Cyanoquinazolines. Molecules 2022, 27, 7156, doi:10.3390/molecules27217156.]. The carefully study is under progress. 

+ 3. In section 2.4, the explanation of Figure 6 (a) is not clear. In addition, it is impossible to distinguish whether the maximum fluorescence intensity is 50% water fraction or 80% water fraction, because the picture is not clear enough.

We rewrite the discussion in section 2.4 and explain more carefully the Figure 6. Also, we changed colors of lines to better distinguish the corresponding mixtures.

+ 4. The author should use three-line form regarding to all the Tables.

We uniformed the tables across the article.

+ 5. The mechanization regarding to the detecting TPA should be provided.

By this moment we can suppose that changes in emission intensity (enhancement or quenching) of com-pound 2e and 2h might be due to intramolecular photo-induced electron transfer (PET) or photo-induced proton transfer [Valeur, B.; Berberan‐Santos, M.N. Molecular Fluorescence; Wiley, 2012; Vol. 53; ISBN 9783527328376.] and the elucidation of exact interaction mechanism of analyte with TFA is under progress.

Round 2

Reviewer 2 Report

I suggest not accept this paper for publication in this journal, should be submitted for a lower impact journal. 

Reviewer 4 Report

I suggest to accept this manuscript for publication.